# HIV infection and cardiovascular disease have both shared and distinct monocyte gene expression features: Women's Interagency HIV study

**Juan Lin** [1], **Erik Ehinger**[2], **David B. Hanna**[1], **Qibin Qi**[1], **Tao Wang**[1], **Yanal Ghosheh**[2], **Karin Mueller**[3], **Kathryn Anastos**[1,4], **Jason M. Lazar**[5], **Wendy J. Mack**[6], **Phyllis C. Tien**[7], **Joan W. Berman**[8], **Mardge H. Cohen**[9], **Igho Ofotokun**[10], **Stephen Gange**[11], **Chenglong Liu**[12], **Sonya L. Heath** [13], **Russell P. Tracy**[14], **Howard N. Hodis**[15], **Alan L. Landay**[16], **Klaus Ley**[2,17☉], **Robert C. Kaplan** [1,18☉]*

1 Department of Epidemiology and Population Health, Albert Einstein College of Medicine, Bronx, NY, United States of America, 2 Department of Inflammation Biology, La Jolla Institute for Allergy and Immunology, La Jolla, CA, United States of America, 3 Department of Cardiology, Eberhard Karls University, Tuebingen University Hospital, Tuebingen, Germany, 4 Department of Medicine, Albert Einstein College of Medicine, Bronx, NY, United States of America, 5 Department of Medicine, Downstate Medical Center, State University of New York, Brooklyn, NY, United States of America, 6 Department of Preventive Medicine, Keck School of Medicine, University of Southern California, Los Angeles, CA, United States of America, 7 Department of Medicine, and Department of Veterans Affairs, Medical Center, University of California, San Francisco, San Francisco, CA, United States of America, 8 Department of Pathology, Albert Einstein College of Medicine, Bronx, NY, United States of America, 9 Department of Medicine, John Stroger Hospital and Rush University, Chicago, IL, United States of America, 10 Department of Medicine, Infectious Disease Division and Grady Health Care System, Emory University School of Medicine, Atlanta, GA, United States of America, 11 Bloomberg School of Public Health, Johns Hopkins University, Baltimore, MD, United States of America, 12 Department of Medicine, Georgetown University Medical Center, Washington, DC, United States of America, 13 Department of Medicine, University of Alabama at Birmingham, Birmingham, AL, United States of America, 14 Department of Pathology & Laboratory Medicine and Biochemistry, University of Vermont Larner College of Medicine, Colchester, VT, United States of America, 15 Department of Medicine, Keck School of Medicine, University of Southern California, Los Angeles, CA, United States of America, 16 Department of Internal Medicine, Rush University Medical Center, Chicago, IL, United States of America, 17 Department of Bioengineering, University of California San Diego, San Diego, CA, United States of America, 18 Fred Hutchinson Cancer Research Center, Division of Public Health Sciences, Seattle, WA, United States of America

☉ These authors contributed equally to this work.
* robert.kaplan@einsteinmed.edu

**Data Availability Statement:** The study data are an open scientific resource that can be used for purposes consistent with study policies and

## Abstract

Persistent inflammation contributes to the development of cardiovascular disease (CVD) as an HIV-associated comorbidity. Innate immune cells such as monocytes are major drivers of inflammation in men and women with HIV. The study objectives are to examine the contribution of circulating non-classical monocytes (NCM, CD14$^{dim}$CD16$^+$) and intermediate monocytes (IM, CD14$^+$CD16$^+$) to the host response to long-term HIV infection and HIV-associated CVD. Women with and without chronic HIV infection (H) were studied. Subclinical CVD (C) was detected as plaques imaged by B-mode carotid artery ultrasound. The study included H-C-, H+C-, H-C+, and H+C+ participants (23 of each, matched on race/ethnicity, age and smoking status), selected from among enrollees in the Women's Interagency HIV Study. We assessed transcriptomic features associated with HIV or CVD alone or

participant informed consent. Researchers can request access through the MACS/WIHS Combined Cohort Study Data Analysis and Coordinating Center (DACC), see the website for additional details: https://www.nhlbi.nih.gov/science/macswihs-combined-cohort-study#how-can-researchers-access-study-data?.

**Funding:** Data in this manuscript were collected by the Women's Interagency HIV Study (WIHS), now the MACS/WIHS Combined Cohort Study (MWCCS). The contents of this publication are solely the responsibility of the authors and do not represent the official views of the National Institutes of Health (NIH). MWCCS (Principal Investigators): Atlanta CRS (Ighovwerha Ofotokun, Anandi Sheth, and Gina Wingood), U01-HL146241; Baltimore CRS (Todd Brown and Joseph Margolick), U01-HL146201; Bronx CRS (Kathryn Anastos and Anjali Sharma), U01-HL146204; Brooklyn CRS (Deborah Gustafson and Tracey Wilson), U01-HL146202; Data Analysis and Coordination Center (Gypsyamber D'Souza, Stephen Gange and Elizabeth Golub), U01-HL146193; Chicago-Cook County CRS (Mardge Cohen and Audrey French), U01-HL146245; Chicago-Northwestern CRS (Steven Wolinsky), U01-HL146240; Northern California CRS (Bradley Aouizerat, Jennifer Price, and Phyllis Tien), U01-HL146242; Los Angeles CRS (Roger Detels and Matthew Mimiaga), U01-HL146333; Metropolitan Washington CRS (Seble Kassaye and Daniel Merenstein), U01-HL146205; Miami CRS (Maria Alcaide, Margaret Fischl, and Deborah Jones), U01-HL146203; Pittsburgh CRS (Jeremy Martinson and Charles Rinaldo), U01-HL146208; UAB-MS CRS (Mirjam-Colette Kempf, Jodie Dionne-Odom, and Deborah Konkle-Parker), U01-HL146192; UNC CRS (Adaora Adimora), U01-HL146194. The MWCCS is funded primarily by the National Heart, Lung, and Blood Institute (NHLBI), with additional co-funding from the Eunice Kennedy Shriver National Institute Of Child Health & Human Development (NICHD), National Institute On Aging (NIA), National Institute Of Dental & Craniofacial Research (NIDCR), National Institute Of Allergy and Infectious Diseases (NIAID), National Institute Of Neurological Disorders And Stroke (NINDS), National Institute Of Mental Health (NIMH), National Institute On Drug Abuse (NIDA), National Institute Of Nursing Research (NINR), National Cancer Institute (NCI), National Institute on Alcohol Abuse and Alcoholism (NIAAA), National Institute on Deafness and Other Communication Disorders (NIDCD), National Institute of Diabetes and Digestive and Kidney Diseases (NIDDK), National Institute on Minority Health and Health Disparities (NIMHD), and in

comorbid HIV/CVD comparing to healthy (H-C-) participants in IM and NCM isolated from peripheral blood mononuclear cells. IM gene expression was little affected by HIV alone or CVD alone. In IM, coexisting HIV and CVD produced a measurable gene transcription signature, which was abolished by lipid-lowering treatment. In NCM, versus non-HIV controls, women with HIV had altered gene expression, irrespective of whether or not they had comorbid CVD. The largest set of differentially expressed genes was found in NCM among women with both HIV and CVD. Genes upregulated in association with HIV included several potential targets of drug therapies, including LAG3 (CD223). In conclusion, circulating monocytes from patients with well controlled HIV infection demonstrate an extensive gene expression signature which may be consistent with the ability of these cells to serve as potential viral reservoirs. Gene transcriptional changes in HIV patients were further magnified in the presence of subclinical CVD.

## Introduction

In people living with HIV, chronic inflammation and immune activation contribute to increased cardiovascular disease (CVD) [1], which by 2030 may affect three-quarters of this patient population [2]. Circulating blood biomarkers of inflammation and coagulation such as C-reactive protein, interleukin (IL)-6, CD14 and D-dimer correlate with atherosclerotic risk [3–7]. However, despite having reproducible associations with disease, these protein biomarkers are blunt instruments to understand the mechanisms and conditions by which HIV gives rise to CVD. Circulating factors, which can be produced by multiple tissues, do not provide cell type-specific information and their blood concentrations can vary depending upon the overall health of the liver, kidney, and other metabolic organs. Thus, the relationship of blood biomarkers with CVD can be confounded by factors that determine protein production, consumption and clearance. An alternative, accessible way to assess risk and mechanisms of disease is transcriptomic analysis of peripheral blood mononuclear cells (PBMCs). The circulating monocyte might be an especially informative sentinel PBMC. In addition to being central to the pathogenesis of atherosclerosis, monocytes are activated as part of the HIV-related systemic inflammatory response and may serve as an HIV reservoir in treated infection [8].

Previously, we used RNA sequencing to study gene expression in circulating classical monocytes (CM) among persons with chronic HIV infection and/or atherosclerotic CVD [9]. In CM, defined as CD14+CD16- cells, both of these disease conditions were associated with a transcription signature of ~200 genes, versus matched disease-free controls. Approximately 20% of the differentially expressed genes (DEGs) were shared between HIV infection and CVD, including suppression of anti-inflammatory *PPAR*, *RXR* and *LXR* gene pathways and upregulated expression of *IL-23*, *IL-6*, *IL-1α* and tissue factor pathway inhibitor-2 (*TFPI-2*). We also identified a beneficial effect of statin-based lipid-lowering treatment (LLT) on CM, such that use of LLT was associated with activated *PPAR* and *LXR*, along with suppression of genes related to the inflammatory cytokine cascade (*IL6*, *TREM1*, *TNF*, *IL1a*, *IL1b*, *IFNg*) and M1-associated transcription factors (*RELA*, *HMGB1*, *FNKB1*).

We applied the same experimental approach to the other two major subsets of monocytes, in order to characterize altered gene expression in the context of HIV infection, CVD and LLT use. Non-classical monocytes (NCM, CD14dimCD16+) are motile cells that patrol the endothelium and extravasate in response to damage [10]. We previously showed that in women with HIV, but not in comparably studied women without HIV infection, CVD was associated with

coordination and alignment with the research priorities of the National Institutes of Health, Office of AIDS Research (OAR). MWCCS data collection is also supported by UL1-TR000004 (UCSF CTSA), UL1-TR003098 (JHU ICTR), UL1-TR001881 (UCLA CTSI), P30-AI-050409 (Atlanta CFAR), P30-AI-073961 (Miami CFAR), P30-AI-050410 (UNC CFAR), P30-AI-027767 (UAB CFAR), and P30-MH-116867 (Miami CHARM). This work is supported by R01-HL148094-01 (R.C. Kaplan & K. Ley), with additional support from R01 HL140976. D.B. Hanna was supported by K01-HL-137557. J.W. Berman was supported by 1R01MH112391. P.C. Tien was supported by the K24 AI 108516. The funders had no role in study design, data collection and analysis, decision to publish, or preparation of the manuscript.

**Competing interests:** The authors have declared that no competing interests exist.

lower cell surface expression of CXCR4 on NCM [11]. Intermediate monocytes (IM, CD14$^+$CD16$^+$) are specialized in antigen presentation [12] but do not have a well-defined function in HIV infection. Greater abundance of IM has been associated with CVD [13, 14]. While prior work has described transcriptomic features of circulating monocytes in HIV-1 infection [15] or CVD [16], the HIV- and CVD-associated gene expression profiles of NCM and IM have not been well described in joint fashion.

## Materials and methods

### Source population and participants' selection

The Women's Interagency HIV Study (WIHS) is a prospective multicenter cohort study of women with or at risk for HIV infection, which was initiated in 1994 and features semiannual core study visits [17]. An unselected sample of enrollees underwent high-resolution B-mode carotid artery ultrasound to image six locations in the right carotid artery [18]. Standardized procedures were used to image the near and far walls of the common carotid artery, carotid artery bifurcation, and internal carotid artery. All participants provided informed written consent and Institutional Review Board approval was obtained by each participating institution, with the study being compliant with the Declaration of Helsinki principles.

To study the independent and joint associations of HIV infection and CVD with gene expression of IM and NCM, comparisons across frequency-matched groups were performed. Using information from multiple data collection episodes (S1 Fig), participants were selected as four groups based on HIV serostatus and subclinical CVD status. CVD was defined as the presence of carotid artery plaque evidenced by an area with localized IMT >1.5 mm in at least one of the six imaged artery locations. The groups included those with comorbid HIV and CVD (H+C+), those with HIV alone (H+C-), those with CVD alone (H-C+), and those with neither HIV nor CVD (H-C-). Women who used LLT (in almost all instances, including a statin) were excluded from the C- groups. We created 23 quartets, each containing one participant from each of the four HIV/CVD groups. Within each quartet, participants were matched as closely as possible by race/ethnicity, age (within 5 years), smoking history, and calendar date (within 1 year). A total of 92 participants were selected from among 1,865 participants of the WIHS who had vascular imaging data. After exclusion of women with insufficient or low-quality RNA or inadequate availability of sample, there remained 87 in our IM study (22 H+C+, 22 H+C-, 20 H-C+, and 23 H-C-) and 88 in our NCM study (22 H+C+, 22 H+C-, 21 H-C+, and 23 H-C-).

### PBMC thawing and monocyte isolation

At each WIHS visit, PBMCs were isolated from venous blood by standardized methods, frozen, and stored in liquid nitrogen in a specimen repository. Cryopreserved PBMCs stored in liquid nitrogen were warmed in 37°C water bath, then removed and immediately diluted with 1 mL of warm cRPMI and transferred and diluted again in an additional 8mL of warm cRPMI. Samples were then washed with PBS and stained with viability reagent (Ghost Dye$^{TM}$ Red 710, Tonbo Bioscience) according to manufacturer's recommendation. Samples were stained with antibody cocktail containing CD14 (M5E2, Biolegend), CD16 (3G8, Biolegend), and dump channel: CD3 (OKT3, Tonbo Bioscience), CD19 (HIB19, Tonbo Bioscience), CD56 (5.1H11, Biolegend), CD66b (G10F5, Biolegend), gated based on living cells and excluding dump channel-positive cells. Intermediate monocytes were defined as CD14$^+$CD16$^+$ and non-classical monocytes were defined as CD14$^{dim}$CD16$^+$. Monocytes were sorted directly into TRIzol® LS Reagent (Life Technologies) and frozen at -80°C. FCS files were exported from FACS Diva (BD Bioscience) and processed using FlowJo v10.2 (FlowJo, LLC). The quantification of

intermediate and non-classical monocytes are estimated from the FACS data from all PBMC samples.

## RNA isolation and sequencing

All sorted samples frozen at -80°C were thawed to room temperature together and processed in one batch. A custom script on a Beckman Coulter Biomek FXP was used to extract total RNA from Trizol using the Direct-zol 96 MagBead RNA kit (Zymo, R2100). Ribosomal RNA was depleted using Ribo-Zero rRNA Removal Kit (Illumina, MRZH11124). 100 ng of each sample's RNA was then prepared into sequencing libraries, according to manufacturer's instructions, using the Truseq Stranded Total RNA Library Prep Kit (Illumina, RC-122-2203). The resulting libraries were deep sequenced on the Illumina HiSeq 4000, using single-end reads with lengths of 50 nucleotides. The single end, 50-bp RNA-Seq reads that passed Illumina filters, were filtered for reads aligning to tRNA, rRNA, and adapter sequences before RNA metrics were calculated. The reads were then aligned to human genome version hg19 (GRCh37.p13 (https://www.gencodegenes.org/releases/19.html)) using TopHat v1.4.1. Post mapping QC was conducted with RSeQC package for quality parameters such as read quality, read distribution, junction saturation, junction annotation and gene body coverage.

## Statistical analysis

Demographic and behavioral variables, HIV-related risk factors, cardiometabolic risk factors, as well as serum inflammation and innate immunity markers, which were measured based on the core study visit closest to the PBMC visit, were compared across four participant groups.

The RNA sequencing generated, on average, 75 million reads per sample. Raw reads were mapped to 58,396 annotated Ensemble coding transcripts using Tophat v2. Noisy, low-expression transcripts with maximum raw counts less than 10 across samples were filtered out to increase the sensitivity of the analysis. Differential expression (DE) analysis was performed for three contrasts, comparing those with HIV alone, CVD alone, or comorbid HIV and CVD, versus the comparison group of H-C- women (**Fig 1A**). To remove the confounding effects of LLT use, another two contrasts compared those free of LLT (H+C+LLT-) and LLT users(H+C +LLT+) versus H-C- were conducted. DE analysis was only performed on protein coding genes that passed the filtering using the R package "DESeq2" [19]. The statistical model was additionally adjusted by the matching unit (e.g., quartet). To eliminate the effect of large fold changes due to low counts or high dispersion on DEG ranking, the "apeglm" method was used for effect size shrinkage [20]. For a given pairwise comparison, a DEG was recognized as a gene with FDR adjusted p-value less than 0.05 and log fold change (LFC) greater than 1 or less than -1. Heatmaps of DEGs of three comparisons were generated using the R package "ComplexHeatmap" [21] to visualize the expression profiles of all DEGs. Venn diagrams showed the relationships of gene signatures.

Fold change and adjusted p-values related to each DEG in pair wise comparisons were then uploaded into Qiagen's system for Ingenuity Pathway Analysis (IPA). Core analysis of IPA was subsequently performed and pathways and biofunctions with p-value≤0.05 were recognized as significant. Canonical pathways, diseases and functions that are most significantly enriched in each gene signature were identified.

The whole gene sets for IM and NCM were screened and filtered to remove low quality genes and outlier samples to improve the accuracy of network construction. A weighted gene co-expression network analysis (WGCNA) was constructed using the R package "WGCNA". For each module identified in IM and NCM, the expression profiles of module genes were summarized as the module eigenvalue (ME). Differences in MEs among HIV/CVD groups

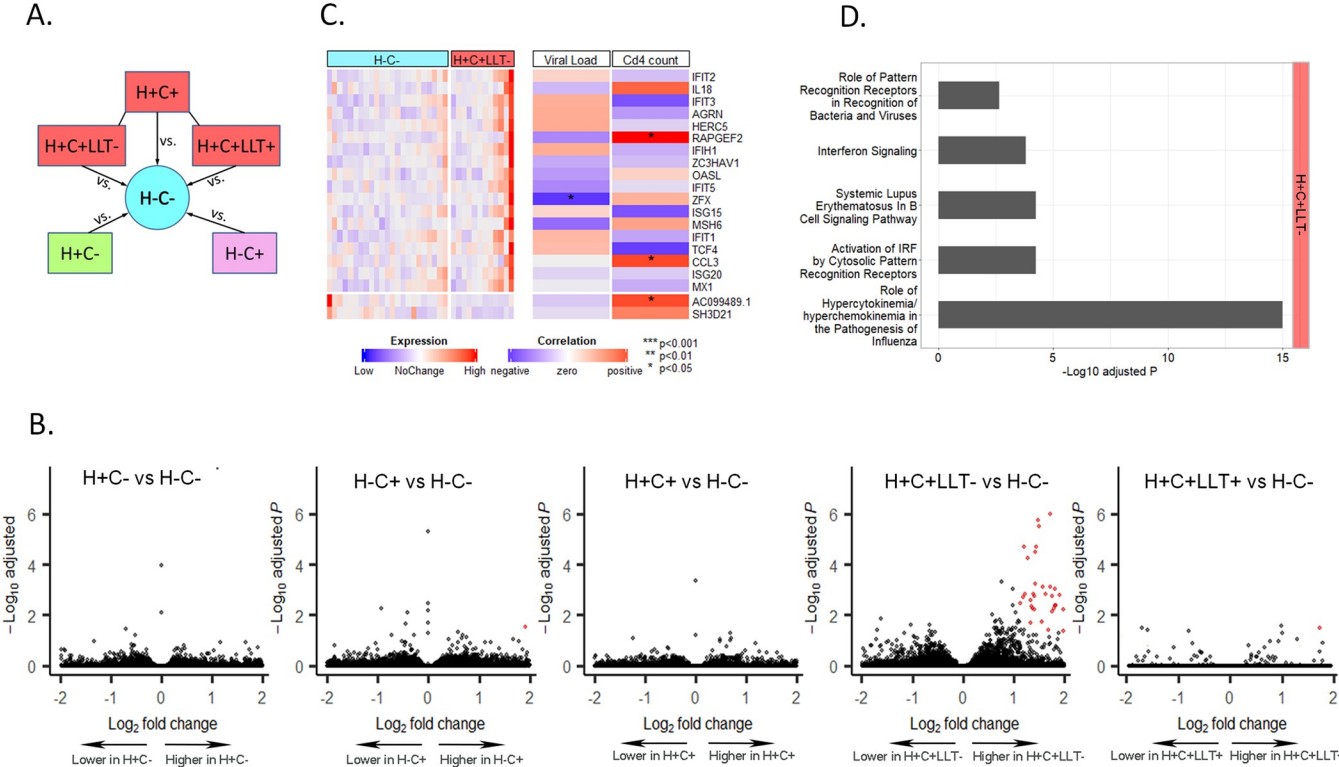

**Fig 1. Results for intermediate monocyte analysis. A.** Schematic of study design showing five comparisons. HIV(H) + or—designates presence or absence of HIV infection. CVD(C) + or—designates of absence of cardiovascular disease. C+ groups were further stratified according to lipid lowering treatment (LLT) use. **B.** Volcano plots for differential gene expression analysis in comparisons of H+C-, H-C+ and H+C+ versus H-C-, as well as comparisons of H+C +LLT- and H+C+LLT+ versus H-C-. Red genes indicate significance by FDR < 0.05 and |log$_2$ fold-change|>1(**S2 Table**). **C**. Heatmap showing expression of top up- and down-regulated differentially expressed genes(DEGs) found in women with H+C+LLT- versus H-C- and their correlation with plasma HIV RNA (Viral load) and CD4+ T cell count (CD4 count). **D.** Top significantly activated pathways predicted by the fold change of DEGs found in H+C+LLT- vs. H-C-comparison (**S2 Table**), using ingenuity pathway analysis.

were compared via one-way ANOVA analysis. P-values were adjusted by the Benjamini-Hochberg (BH) method. Correlations between MEs and HIV- or CVD-related clinical traits were also determined. Next, genes of specific target modules in IM and NCM were subjected to Gene Ontology (GO) term assignment, as well as other ontology enrichment analyses from the Kyoto Encyclopedia of Genes and Genomes (KEGG), the Human Protein Atlas (HPA), and the Comprehensive Resource of mammalian protein complexes (CORUM) databases, using the R package "gprofiler".

All analyses were performed using SAS 9.4 (SAS Institute Inc., Cary, North Carolina, USA) and R 3.5.3 (R Project for Statistical Computing, Geneva).

## Results

A total of 92 participants were selected as four groups based on HIV and subclinical CVD status, as defined by presence of carotid artery plaque measured by B mode ultrasound. The groups included those with comorbid HIV and CVD (H+C+), HIV alone (H+C-) or CVD alone (H-C+), and those with neither HIV nor CVD (H-C-).

Participants were 45 years old (median) and 95% Black or Hispanic (**S1 Table**). Among women in the H+C+ and H+C- groups, over 84% used HAART, and 59% had undetectable plasma HIV-1 RNA level; among these 44 women with HIV, only 14% had HIV RNA above

1000 copies/ml. The most pronounced clinical difference among the groups was LLT, which was used by about one-quarter of H-C+ women and over two-fifths of H+C+ women, while none of the H-C- or H+C- women used LLT.

Cell counts of IM and NCM were not significantly different across H+C+, H+C-, H-C + and H-C- groups (*P* = 0.546 and 0.522, **S2 Fig**). After filtering low-expression transcripts, 18,505 and 18,670 protein-coding genes remained for analysis of IM and NCM, respectively. Differential expression analysis of three contrasts, comparing those with HIV alone, CVD alone, or comorbid HIV and CVD, versus the comparison group of H-C- women, were performed (**Fig 1A**).

## Analyses of intermediate monocytes

For HIV infection alone (H+C-), as compared with H-C- comparators, no DEGs in IM were found (**Fig 1B**). Among women with CVD alone (H-C+ versus H-C-), *NAIP* was the only DEG found in IM (**Fig 1B**). Further analyses divided the H-C+ population into LLT+ and LLT- groups, and compared them with H-C- group. This analysis showed a weak gene expression signature for CVD alone in HIV-uninfected women, and little difference between those using LLT and those not using LLT (**S2 Table**).

By contrast, we detected an extensive DEG signature in IM associated with non-LLT users of comorbid HIV/CVD participants (H+C+ LLT- versus H-C-, **Fig 1B**). The signature of H+C + differed depending on whether or not LLT was used. H+C+ LLT users (N = 10) had only one DEG in IM versus H-C- (*HSPD1*). In contrast, H+C+ women who were not using LLT (N = 12) had a signature of 59 DEGs in IM (57/59 up-regulated), versus H-C- (**S2 Table**). Up-regulated DEGs included known atherosclerosis genes such as the liver X receptor gene *NR1H2* [22], and *NEXN, TRAF1* [23], *TLR7 and LGALS3BP* [24], implicating tumor necrosis factor and toll-like receptor signaling and glycan-binding protein pathways. Many upregulated genes in the H+C+ signature (non-LLT users) have known roles in HIV infection, including several innate immune response genes previously shown to be stimulated by HIV *in vitro* (*IFIT1/2/3, DDX58, MX1*) [25], *CCL3* and *CCL4L2*, which encode macrophage inflammatory protein-1 alpha and beta [26], complement factor B (*CFB*) of the alternative complement pathway [27, 28], interleukins/ interleukin receptors (*IL7, IL18, IL15RA*) and the interferon-stimulated genes e.g., *ISG15, ISG20, USP18* [29]. Expression levels of top H+C+ associated genes were not consistently correlated with plasma HIV RNA or CD4+ T-cell count (**Fig 1C**). Only 1 (1.7%) and 11(18.6%) of 59 IM DEGs were correlated with HIV RNA and CD4+ T cell count at *P*<0.05, respectively. These suggested that altered regulation of these DEGs was not driven by known HIV immunologic or virologic parameters.

Among the comorbid HIV and CVD group, further analysis comparing LLT users with non-LLT users revealed a 22-gene signature associated with LLT. Eighteen of them were down-regulated and 4 were up-regulated in LLT users (**S3A and S3B Fig**). The role of these 22 LLT-associated DE genes in IM was further investigated by ingenuity pathway analysis (IPA). Canonical pathways such as "Recognition of Bacteria and Viruses", "Interferon Signaling" and "Interferon Induction and Antiviral Response" were found significantly enriched among these DEGs, suggesting that LLT had an anti-inflammation potential in IM of persons with H+C + status (**S3C Fig**).

IPA was conducted based upon 59 DEGs associated with the H+C+LLT- group (**S2 Table**) to characterize the gene transcription signal associated with comorbid HIV and CVD. The top five significantly activated canonical pathways are illustrated in **Fig 1D**. These top pathways are mainly related to pathogen recognition, signal transduction and cellular immune response. Key genes encoding messenger molecules and RNA helicases, like *IL18, CCL3, CXCL10, IFIH1*

and *DDX58*, played an important role in these pathways. IPA analysis also revealed 19 unique disease and disorder functions that were significantly associated with these DEGs, the top five of which related to inflammatory response and immunological disease (**S3A Table**).

In addition, gene co-expression networks were built in IM by Weighted Gene Co-expression Network Analysis (WGCNA). This resulted in the detection of 8 gene modules, but no significant difference in module eigenvalues was observed in any of these IM modules across the four study groups.

In summary, among women with comorbid HIV and CVD, we observed a panoply of up-regulated genes in IM, which was extinguished by LLT use. IPA indicated that many of these DEGs are involved in cellular immune responses. In those with HIV infection, presence versus absence of LLT was associated with reduction in inflammatory and innate immune regulatory signals, whereas LLT effects on gene expression were muted among those without HIV. Neither HIV alone nor CVD alone produced a distinguishing IM gene transcription signature.

## Analyses of non-classical monocytes

When compared with H-C- controls, women with HIV had altered gene expression in NCM irrespective of whether or not they had comorbid CVD. As was found for IM, in NCM the H+C+ group had the largest set of DEGs (versus H-C-, 90 DEGs, 64 up-regulated and 26 down-regulated, **Fig 2A and 2B**, **S4 Table**). Up-regulated DEGs of this gene signature included T-cell development, growth and proliferation genes (*BATF*, *CCL5*, *CD2*, *EOMES*) [30] and *F2R*, a recognized genetic locus influencing clinical CVD [31] and atherosclerosis development [32]. Several cell adhesion and migration related genes were down-regulated (*ADAMTS1*, *ADGRG3*, *C5AR2*, *CSF3R*). A smaller, partially overlapping set of 37 DEGs (35 up-regulated) was associated with HIV infection alone (H+C- vs H-C-, **Fig 2A and 2B**, **S4 Table**). Up-regulated DEGs for HIV alone involve innate or adaptive immune response (*LAG3*, *CCL5*, *MCOLN2*), T-cell proliferation (*ITGAD*), cell adhesion or migration (*VCAM1*, *NUAK1*, *CCDC141*, *IL12A*, *KIF19*) and signal transduction (*IFNAR1*, *PLEKHG1*) [33]. DEG signatures of the H+C+ and H+C- groups shared 21 genes, all of which were up-regulated in both groups. We next examined the correlation of plasma HIV RNA and CD4+ T-cell count with the expression of top DEGs associated with either H+C+ or H+C- (**Fig 2C and 2D**). NCM signature genes that were up-regulated in H+C+ and H+C- groups were more highly expressed in those with higher plasma HIV RNA and lower CD4+ T-cell. Conversely, down-regulated genes were associated with lower HIV RNA and higher CD4+ T-cell count. In addition, 37% and 32% of NCM DEGs were correlated with HIV RNA and CD4+ T cell count at $P<0.05$, respectively. These suggested that expression of NCM DEGs in the presence of HIV infection were associated with immunological and virologic status.

We conducted IPA of the 90-gene signature of H+C+ and the 37-gene signature of H+C- (**S4 Table**) to identify common or distinct elements (**S3B**, **S3C Table**). Top activated canonical pathways for H+C+ were mainly related to T-lymphocyte signaling (**Fig 2E**). Up-regulation of six DEGs (*GRAP2*, *LCK*, *PPP2R2B*, *BATF*, *EOMES* and *LAG3*) and down-regulation of *RASD1* contribute to the enrichment of these pathways. Top activated pathways in the H+C- signature were related to cytokine production and dysregulation, and major contributing DEGs to these pathways, *CCL5*, *IFNG* and *IL12A*, are all involved in type-1 helper T-cell (Th1) responses. In both H+C+ and H+C-, IPA identified up-regulation of inflammation as the lead disease and disorder while H+C+ additionally implicated immunological disease.

Further analyses examined the influence of LLT on gene transcription in NCM through stratification by LLT use (**S5 Table**). Among 12 H+C+ women who were not using LLT, we identified 25 DEGs versus H-C-. Of these DEGs, 16 (64%) overlapped with the 90 DEGs of the

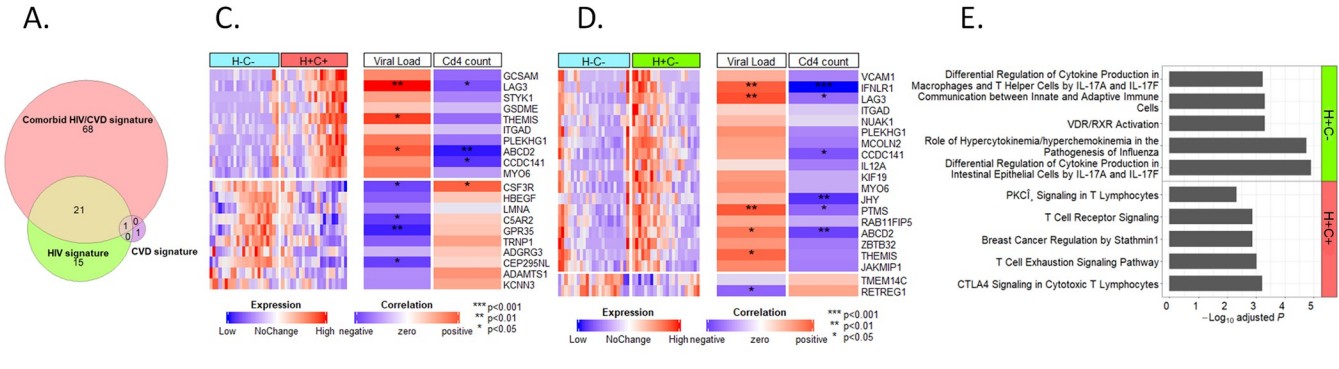

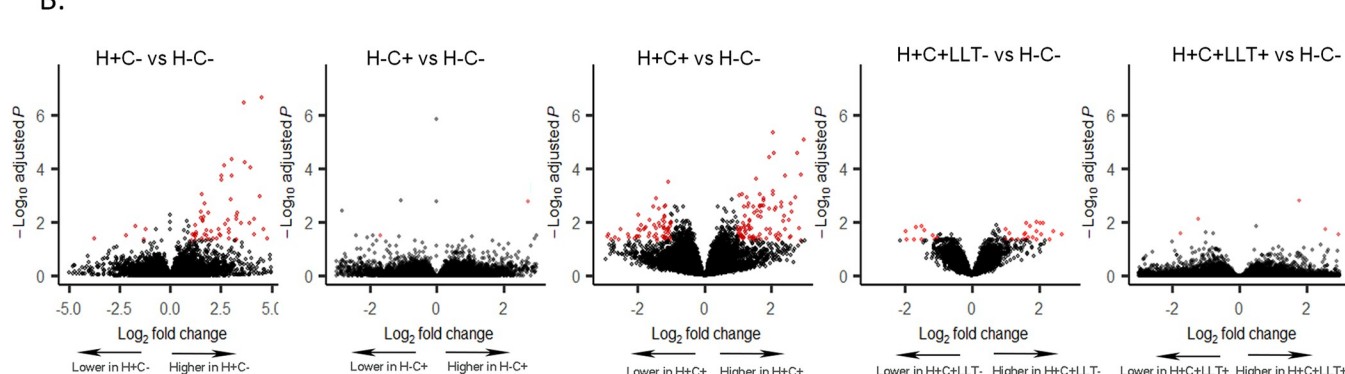

**Fig 2. Results for non-classical monocyte analysis. A.** Venn Diagram of differential expression gene (DEGs) lists for the three major comparisons, all contrasted versus HIV(H)-CVD(C)- control group. **B.** Volcano plots for differential gene expression analysis in comparisons of H+C-, H-C+ and H+C+ versus H-C-, as well as comparisons of H+C+LLT- and H+C+LLT- versus H-C-. Red genes indicate significance by FDR < 0.05 and |log₂ fold-change |>1(**S4 Table**). **C-D.** Heatmap showing expression of top up- and down-regulated DEGs found in women with H+C+ (**C**) and women with H+C-(**D**) vs. H-C- and their correlation with HIV RNA (Viral load) and CD4+ T cell count (CD4 count). **E.** Top significantly activated or suppressed pathways predicted by the fold change of DEGs found in H+C- or H+C+ vs. H-C- comparison (**S4 Table**), using ingenuity pathway analysis.

overall analysis of H+C+ versus H-C-, indicating results from the non-LLT users were consistent with those from the overall H+C+ group. When we compared expression profiles of LLT users versus non-LLT users among those with comorbid HIV and CVD, we identified only 3 DEGs which all were found significantly up-regulated in LLT users (*MTRNR2L12, MTRNR2L1* and *HSPD1*). Thus, little difference in NCM gene expression was found between H+C + women who were not using LLT and who were using LLT.

We also looked for genes with altered expression in NCM in the presence of CVD alone (H-C+ versus H-C-). In this comparison, only two DEGs (*NUAK1 and AQP9*) were identified (**S4 Table**). In summary, in analyses of NCM, the largest gene transcription signature was associated with HIV infection (whether or not CVD was also present), whereas CVD alone had a minimal NCM signature.

WGCNA was performed in NCM and detected 15 gene modules. Compared with H-C- participants, expression of the salmon (color) module in NCM was significantly elevated in both H+C- and H+C+ groups, while expression of the magenta module showed a statistically significant contrast being highest in H-C- and lowest in H+C+ (**Fig 3**). The genes contained in the salmon and magenta modules are listed in **S6 Table**. The salmon module includes a few genes involved in monocyte function (e.g. *SYT11, CRTAM, MYO6*), while the magenta module encodes many genes that involved in metabolism, as well as glycosaminoglycans and glycolipids (e.g. *ADPGK, PDPK1, PRKAG2*).

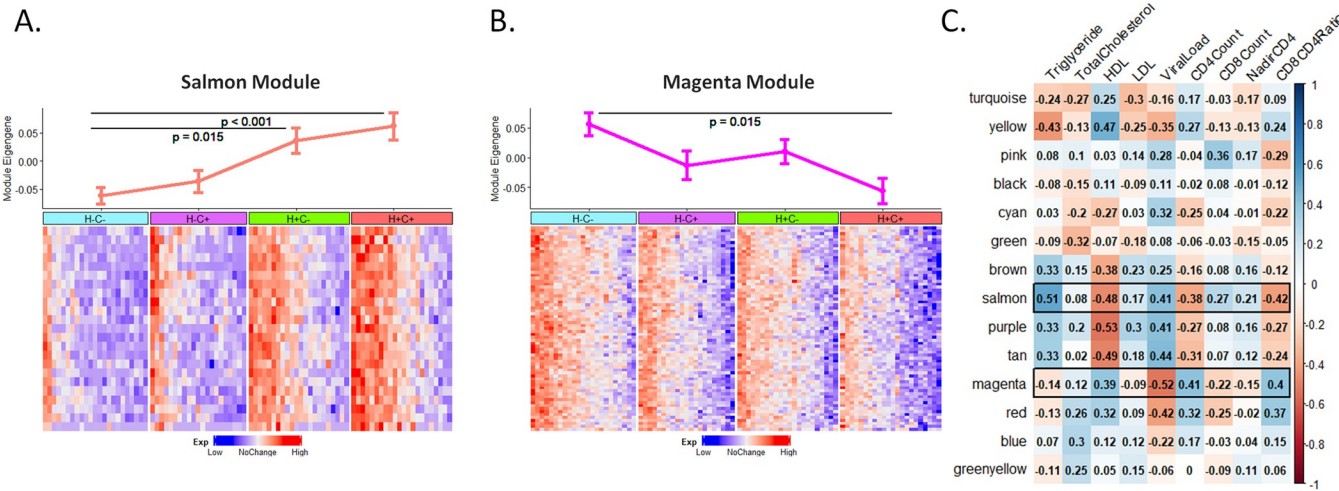

**Fig 3. Weighted gene co-expression network analysis results in non-classical monocyte (NCM). A.** Comparison of module eigenvalues (MEs) of salmon module and expression profiles of 22 salmon module genes in NCM by HIV and CVD status. **B.** Comparison of module eigenvalue of magenta module and expression profiles of 70 magenta module genes in NCM by HIV and CVD status. **C.** Spearman correlation between MEs of all gene modules identified in NCM, and CVD- or HIV-related clinical features.

Module-trait relationships are presented as a correlation matrix between module eigenvalues and clinical features, and the significance of relationships is shown in color (**Fig 3C**). The elevated expression of the salmon module in NCM was significantly related to higher HIV RNA, lower CD4+ T cell count and CD4/CD8 ratio, and lipid alterations typical of HIV infection (increases in triglyceride levels, decreases in HDL-cholesterol levels, and no changes in LDL-cholesterol levels [34]). The associations of magenta module expression with most of these clinical features were also significant, but in an opposite direction of the salmon module. The salmon and magenta modules comprised 22 and 70 genes, respectively. Of 22 salmon genes, 10 (45%) were found in the 37-gene signature of H+C- and 17(77%) were found the 90-gene signature of H+C+. Not much overlap was detected between NCM gene signatures and magenta module. Only 4 out of 70 magenta genes were found in the 90-gene signature of H+C+.

For the salmon module, the enriched Gene Ontology biological function terms in the annotation systems were mostly related to "Regulation of leukocyte activation" (adjusted *p*-value = 0.036) and "Identical protein binding" (adjusted *p*-value = 0.042). A few key genes that contributed to these terms are *SYT11* (encodes synaptostigmin11 which is involved in endocytosis and vesicle recycling), *CRTAM* (mediates heterophilic cell-cell adhesion), *MYO6* (encodes myosin-VI which mediates vesicular membrane trafficking and cell migration) and *CD2* (interacts with lymphocyte function-associated antigen to mediate cell-cell adhesion). For magenta module genes, the top enriched terms in the KEGG pathway databases were "Insulin signaling pathway" (adjusted *p*-value = 0.008) and "Insulin resistance" (adjusted *p*-value = 0.04). Important contributors to the enrichment are *PDPK1* (a serine/threonine kinase which acts as a master kinase in the transduction of signals from insulin), *PRKAG2* (an energy sensor protein kinase that plays a key role in regulating cellular energy metabolism), *RHOQ* (a GTPase that causes the formation of filopodia) and *PPP1CB* (a phosphatase that is involved in the regulation of a variety of cellular processes).

## Discussion

Neither HIV nor CVD was associated with differences in the number or distribution of IM and NCM. Several prior studies in the general (non-HIV) population have associated

**Table 1. Summary of findings relating gene expression in classical, intermediate, and non-classical monocytes with HIV infection and subclinical CVD.**

| | Classical monocytes | Intermediate monocytes | Non-classical monocytes |
|---|---|---|---|
| **HIV alone signature** | Signature of ~200 DEGs | None | Signature of 37 DEGs |
| **CVD alone signature** | Signature of ~200 DEGs. ~20% of the DEGs are shared between HIV alone and CVD alone, suggesting pro-inflammatory and pro-coagulant pathways in common. | Negligible (only one DEG in overall CVD group; three in those who were LLT users; one in those who were non-LLT users). | Negligible (one DEG, NUAK1, a gene that was also found in HIV signature) |
| **Comorbid HIV and CVD signature and effects of lipid lowering treatment (LLT)** | People with both HIV and CVD had a weak DEG signature (9 DEGs), less than expected from the sum of the two disease conditions. Further analyses accounting for LLT use found that HIV/CVD group who were non-LLT users had a 108 DEG signature. Meanwhile, LLT use abolished the gene expression signature in patients with comorbid HIV/CVD. | Only one DEG was found (HSPD1), for members of the HIV/CVD comorbid group who were using LLT. In contrast, those not using LLT had a signature of 59 (mostly upregulated) DEGs. Although these DEGs include many genes with known roles in HIV infection, their expression levels were not associated with plasma HIV RNA or CD4+ T cell count. | In people with both HIV and CVD, a 90-DEG signature was found, which included many of the DEGs associated with HIV alone. The degree of upregulation or downregulation of DEGs in the HIV alone or HIV/CVD comorbid group was correlated with plasma HIV RNA and CD4+ T cell count. Further analyses limited to non-LLT users produced an overlapping set of DEGs as the overall analysis of the HIV/CVD group. |
| **Interpretation** | CVD and HIV switch on many of the same pro-inflammatory genes in CM. LLT reduces the CM gene expression response to baseline, such that the CVD signature of CM can only be detected in LLT-free patients | IM gene expression is little affected by HIV or CVD alone. In IM, coexisting HIV and CVD produces a measurable gene transcription signature, which is abolished by LLT. The DEGs implicated are involved in pathways of HIV disease, although their expression is not associated with HIV clinical immunologic or virologic status. | NCM is little affected by subclinical CVD alone. In NCM, HIV has a gene-upregulatory effect that correlates with HIV clinical status variables including viral load and CD4+ T cell count. The co-occurrence of CVD adds more genes to the proinflammatory gene signature of HIV. Use of LLTs dampens the pro-inflammatory signal of HIV/CVD. |
| **Reference** | Ehinger et al, *Cardiovasc Res* 2021, PMID: 32658258 | Present study | Present study |

increased frequency of the CD14+ (particularly IM) cell population with clinical CVD events and plaque vulnerability [35, 36]. Studies on monocyte subtype distributions in patients with CVD and HIV have been conflicting [37, 38], although expansions of both NCM and IM have been described in a range of viral, autoimmune and inflammatory conditions [39]. Our data support the premise that shifts in gene transcription may be more important to HIV-related disease than changes in relative abundance of monocyte subsets.

Using RNA sequencing, we identified patterns of gene expression in circulating monocytes from women with HIV infection, subclinical CVD, or both conditions. This report using PBMC-derived IM and NCM complements our recent study of CM [9], and the combined data illustrates the relation of each monocyte subset with these clinical states (**Table 1**). Women who survive long-term HIV infection, even while on effective HAART, have changes in monocyte gene expression which may contribute to CVD complications of HIV infection.

First, HIV infection was associated with gene expression changes in NCM. Many of the affected genes are known to be involved in atherosclerosis or response to viral infection. In contrast, gene expression in IM was not altered with HIV infection *per se.*

Second, altered IM gene expression was confined to the group with both HIV and CVD. The IM DEGs associated with the HIV/CVD comorbid condition were not correlated with markers of HIV disease stage including plasma HIV RNA or CD4+ T cell count; this observation is important because it links altered IM gene expression with the presence of HIV-associated CVD, rather than with parameters reflecting HIV disease control.

Moreover, IM gene expression did not vary with the presence or absence of CVD in persons who were not infected with HIV. Thus, while IMs are recognized as a cell population involved in atherosclerosis [13, 14], the role of IMs may be especially important in HIV-associated mechanisms of CVD. In contrast with IMs, NCMs showed little evidence of association with CVD regardless of HIV status. Instead, DEGs we identified in NCM appeared to be

attributable to HIV disease. NCM gene expression levels were found both to differ between HIV versus non-HIV groups as well as to correlate with immunological and virologic status.

In NCM, we were able to identify two gene modules that are associated with HIV and CVD in distinct ways. The salmon module, significantly up-regulated in women with HIV, includes genes involved in cell-cell adhesion, vesicular membrane trafficking and cell migration, which is consistent with the mobile function of NCM and suggests an increased capacity of NCM to patrol in the context of chronic HIV infection. The magenta module, significantly down-regulated in the HIV/CVD comorbid group, encompasses many genes involved in insulin signal transduction, metabolism, as well as glycosaminoglycans and glycolipids. The interactions between endothelial cells and monocytes are controlled by adhesion molecules that are in turn regulated by cellular glycosylation [40]. Down-regulation of glycosylation-related genes may lead to a reduction of post-translational modification of cell-surface adhesion molecules, which might in turn modulate the endothelial-monocyte interactions and mediate an anti-inflammatory effect of NCM.

Another clear finding is that inflammatory gene expression in all three monocyte populations (classical [9], intermediate and non-classical) appeared to be dampened by LLT. This suggests that the putative favorable immunological and anti-atherosclerotic effects of LLT may involve CM, IM and NCM. Forthcoming evidence from the Randomized Trial to Prevent Vascular Events in HIV study will provide guidance relating to the incorporation of statins as part of medical treatment for HIV [41].

The design of this study enabled us to contrast gene signatures of HIV and CVD because an identical study protocol and sample size was used for all participant strata. Women were mostly receiving state-of-the art treatment for HIV infection, which ensures generalizability of results in the contemporary era of effective HAART. Limitations include the cross-sectional nature of the study, the absence of male participants, and the potential for confounding variables to have influenced results, although we used a matched design as a rigorous way to control confounding. Future studies will be needed to elucidate how HIV-related factors, including the amount of HIV viral load, duration of ART and stage of HIV disease at diagnosis, etc., affect the differential gene expression of monocytes. In addition, including plaque stability in determination of subclinical CVD status will provide valuable information for plaque risk stratification.

Pathways and genes we identified to be up-regulated by HIV infection in monocytes may include candidates for intervention to reduce complications of HIV infection. LAG3 (CD223), a strongly up-regulated gene in NCM in both the H+C- and H+C+ conditions, is involved in antigen presentation by dendritic cells to T lymphocytes and is a checkpoint inhibitor target for investigative cancer and immunotherapy drugs [42, 43], while also being implicated in atherosclerotic disease [44]. Thus, leveraging the increased information available from RNA sequencing over candidate gene approaches [45], the study provides several new avenues for investigation the role of monocytes in cardiovascular complications of HIV infection.

## Supporting information

**S1 Fig. Timeline of the study recruitment, data collection and sampling.** The Women's Interagency HIV Study (WIHS) was initiated in 1994 and at the time of this study comprised six U.S. sites [17]. Recruitment in the WIHS occurred in two waves (1994–1995, 2001–2002) from HIV primary care clinics, hospital-based programs, community outreach, support groups, and other locations. The WIHS protocol involves semi-annual follow-up visits with detailed examinations, specimen collection, and structured interviews. Incorporated into our definition of cardiovascular disease were data from two waves of B-mode carotid artery

ultrasound scans conducted approximately 7 years apart, first during 2004–2006 (Carotid ultrasound substudy scan #1) and again during 2010–2013 (Carotid ultrasound substudy scan #2). For the present report, from both the HIV infected and HIV-uninfected participants of WIHS, we selected groups both with and without subclinical cardiovascular disease (C); the C + group was defined as those having at least 1 carotid artery plaque assessed at either vascular scan and the C- group was defined as no carotid artery plaque and either of the two scans. Women who used statin drugs for cholesterol lowering were excluded from the C- groups. We withdrew from the repository a sample of peripheral blood mononuclear cells (PBMCs) that had been collected from the semi-annual WIHS core visit that occurred as close as possible in time to the most recent vascular substudy visit. This study is based on transcriptomes of non-classical monocytes isolated from frozen peripheral blood mononuclear cells (PBMCs) obtained from 88 WIHS participants at WIHS Visits 27 to 36 (corresponding to calendar years 2008 to 2012). If samples were not available or had adequate volume, an alternate was selected from an alternate visit before or after the carotid artery ultrasound scan.
(TIF)

**S2 Fig. Frequency of intermediate and non-classical monocytes to the total monocytes by HIV (H) and CVD(C) status.**
(TIF)

**S3 Fig. Results for comparison between lipid lowering treatment (LLT) users and non-LLT users in intermediate monocytes analysis. A.** Volcano plots for differential gene expression analysis in comparisons of H+C+LLT+ versus H+C+LLT-. Red genes indicate significance by FDR < 0.05 and |log$_2$ fold-change|>1. **B.** Heatmap showing expression of top up- and down-regulated DEGs found in women with H+C+LLT+ versus H+C+LLT-. **C.** Top significantly enriched pathways predicted by the fold change of DEGs found in H+C+LLT+ vs. H+C+LLT- comparison, using ingenuity pathway analysis.
(TIF)

**S1 Table. Characteristics of non-classical monocytes study participants, by HIV infection (H) and subclinical cardiovascular disease (C) status.**
(DOCX)

**S2 Table. Differentially expressed genes in Intermediate monocytes, comparing groups with cardiovascular disease (C+) with or without HIV infection (H) and lipid lowering treatment (LLT), versus H-C- controls.**
(DOCX)

**S3 Table.** Ingenuity pathway analysis core analysis results for differentially expressed genes in intermediate monocytes related to **A.** H+C+LLT- and non-classical monocytes related to **B.** H +C- **C.** H+C+.
(DOCX)

**S4 Table. Log fold change of differentially expressed genes in non-classical monocytes associated with HIV alone, CVD alone and comorbid HIV/CVD.**
(DOCX)

**S5 Table. Differentially expressed genes in non-classical monocytes among women with subclinical cardiovascular disease (C), stratified by HIV infection (H) and lipid-lowering treatment (LLT) status.**
(DOCX)

**S6 Table. Gene names of 22 salmon module genes and 70 Magenta module genes in non-classical monocytes.**
(DOCX)

## Author Contributions

**Conceptualization:** Alan L. Landay, Klaus Ley, Robert C. Kaplan.

**Data curation:** Kathryn Anastos, Jason M. Lazar, Wendy J. Mack, Phyllis C. Tien, Mardge H. Cohen, Igho Ofotokun, Stephen Gange, Chenglong Liu, Sonya L. Heath, Russell P. Tracy, Howard N. Hodis.

**Formal analysis:** Juan Lin, Erik Ehinger, Yanal Ghosheh, Karin Mueller.

**Funding acquisition:** Klaus Ley, Robert C. Kaplan.

**Methodology:** Juan Lin, David B. Hanna, Qibin Qi, Tao Wang, Klaus Ley, Robert C. Kaplan.

**Writing – original draft:** Juan Lin, Klaus Ley, Robert C. Kaplan.

**Writing – review & editing:** Juan Lin, Erik Ehinger, David B. Hanna, Qibin Qi, Tao Wang, Yanal Ghosheh, Karin Mueller, Kathryn Anastos, Jason M. Lazar, Wendy J. Mack, Phyllis C. Tien, Joan W. Berman, Mardge H. Cohen, Igho Ofotokun, Stephen Gange, Chenglong Liu, Sonya L. Heath, Russell P. Tracy, Howard N. Hodis, Alan L. Landay, Klaus Ley, Robert C. Kaplan.

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
