## [Decision Letter · Decision Letter 0]

6 Mar 2023

PONE-D-23-02143HIV infection and cardiovascular disease have both shared and distinct monocyte gene expression features: Women's Interagency HIV StudyPLOS ONE

Dear Dr. Kaplan,

Thank you for submitting your manuscript to PLOS ONE. After careful consideration, we feel that it has merit but does not fully meet PLOS ONE’s publication criteria as it currently stands yet. Therefore, we invite you to submit a revised version of the manuscript that addresses the points raised during the review process in point-to point fashion. Given that these points are adequately addressed, this paper will likely achieve the priority necessary for publication.

We look forward to receiving your revised manuscript.

Kind regards,

Andreas Zirlik, MD

Academic Editor

PLOS ONE

Journal Requirements:

2.We note that the grant information you provided in the ‘Funding Information’ and ‘Financial Disclosure’ sections do not match. 

Reviewers' comments:

Reviewer's Responses to Questions

**Comments to the Author**

1. Is the manuscript technically sound, and do the data support the conclusions?

Reviewer #1: Yes

Reviewer #2: Yes

2. Has the statistical analysis been performed appropriately and rigorously? 

Reviewer #1: Yes

Reviewer #2: Yes

3. Have the authors made all data underlying the findings in their manuscript fully available?

Reviewer #1: No

Reviewer #2: Yes

4. Is the manuscript presented in an intelligible fashion and written in standard English?

Reviewer #1: Yes

Reviewer #2: Yes

5. Review Comments to the Author

Reviewer #1: Major:

• Current analyses in the manuscript focus on peripheral transcriptomes, but their contribution to atherosclerosis remain unclear. It would be helpful to correlate the found signatures to publicly available transcriptomes of stable and unstable carotid plaques. Are the found signatures associating more or less with either one of the plaque phenotypes?

• Down this same track: Are the infected peripheral monocyte closer to monocytes in the plaque. The authors may use available monocyte transcriptomes from available scRNAseq studies.

• DE genes used as input for the pathway analysis are not evident from text or figures. These should be shown. Authors should label the corresponding genes in the volcano plots and explicitly state which genes were included in the pathway analysis, best as supplementary data file.

• E.g. Line 271 on page 7: "The role of these genes in IM was further investigated by ingenuity pathway analysis (IPA)..." and Line 277 page 7: "IPA was conducted to 59 DEGs associated with the H+C+LLT- group to characterize the gene transcription signal associated with comorbid HIV and CVD..." Which genes were used?

• Line 250 on page 7: „By contrast, we detected an extensive DEG signature in IM associated with comorbid HIV/CVD (H+C+ versus H-C-, Fig 1B).“ In Figure 1B, no significantly regulated DEGs are shown for the comparison H+C+ vs H-C-. Line 299 on page 8: „As was found for IM, in NCM the H+C+ group had the largest set of DEGs….”. Please provide a list of genes.

Minor:

• The legends for the Volcano plots in Figures 1 & 2 are missing. I assume red dots are genes that show significant regulation.

• From the Methods section it becomes clear, that the authors performed DE Gene analysis in five comparisons including 6 groups (H+C-, H-C+, H+C+, H+C+LLT-, H+C+LLT+, H-C-). From this perspective, I find the schematic representation in Fig1a misleading, as only 4 groups are depicted. In addition, the orientation of the arrows towards the H-C- group is unclear. What is the intention of the authors?

• In Figure 1b, some significantly regulated genes in the volcano plots are named while others are not (compare volcano plots for H+C+LLT- vs H-C- to H+C+LLT-+vs H-C-). Why were these genes labelled and others not? Presentation of data should be consistent.

• In line with the previous comment: l. 254 on page 7: "Up-regulated DEGs included known atherosclerosis genes such as the liver X receptor gene NR1H222, and NEXN, TRAF123, TLR7 and LGALS3BP24....". Please labele in the volcano plot.

• The pathways shown in Figure 1D are not clearly associated with a particular group. Therefore, the authors are asked to add this information.

• I understand that DE genes with an adjusted p-value <0.05 were included in the pathway analysis. Were significant pathways detected by the adjusted or unadjusted p value? This is not clearly stated in the methods section. Unadjusted p-values are listed in Table S3. The authors are kindly requested to clarify this.

• Please label Figure 3A&B with the corresponding models and refer to them correctly in the text (see line 359, page 9).

Reviewer #2: This manuscript explores the difference in single cell transcriptome of Non Classical Monocytes and Intermediate Monocytes in PBMCs obtained from patient with HIV and CVD. The aim was to identified conserved differentially expressed gene signature in 92 patient divided in four groups according to their diagnosis (H-C- as a control, H+C-, H-,C+ and comorbidity of H+C+). This study is interesting because cardiovascular disease is one of the leading causes of mortality among people living with HIV and the use of PBMCs would be an affordable way to assess the risk and the mechanisms by which HIV engender CVD.

Comments:

The authors indicated that no difference in cell count was found, does this number refer to the quantification of the cells isolated from the CPT tube or the cells they isolated for sequencing? If the latter, can it really be said that there is no difference?

In the results section, second paragraph. The values at the beginning are being presented in percentage, please keep it, it will be easy to correlate the 59% with undetectable RNA and 6 patients with higher RNA.

Do the authors looked at whether the 59% of HIV-positive patients who did not have RNA in plasma have any difference with patients with detected RNA and CDV? Wouldn't the presence or absence of RNA affect the behavior of peripheral cells, in this case monocytes? Or even if the disease is at a very advanced stage. This could perhaps give an idea why in the heatmap of the groups with IHV+ are not so homogeneous.

Heine GH, Rogacev KS had previously reported that the number of intermediate monocytes may be a significant predictor of cardiovascular events. What could be attributed in this case to the no difference found in monocytes between patients with CVD and negative for HIV?

6. PLOS authors have the option to publish the peer review history of their article (what does this mean?). If published, this will include your full peer review and any attached files.

Reviewer #1: No

Reviewer #2: No

---

## [Author Response · Author response to Decision Letter 0]

17 Apr 2023

Response to Reviewer #1: 

Major:

• Current analyses in the manuscript focus on peripheral transcriptomes, but their contribution to atherosclerosis remains unclear. It would be helpful to correlate the found signatures to publicly available transcriptomes of stable and unstable carotid plaques. Are the found signatures associating more or less with either one of the plaque phenotypes?

While the reviewer makes an excellent point, the distinction between stable and unstable plaques is beyond the capability of our vascular imaging method. Subclinical CVD status in our study is determined from carotid artery intima media thickness (CIMT) though B-mode ultrasound images. Although CIMT has been widely used and is the most valid determinant of subclinical atherosclerosis, it does not distinguish between stable and unstable plaques. 

We have included this limitation in the discussion section.

Line 478: …In addition, including plaque stability in determination of subclinical CVD status will provide valuable information for plaque risk stratification. 

Many transcriptomics studies have investigated the gene expression regulation in the progression of unstable plaques using bulk RNA-Seq. However, carotid plaques are characterized by heterogeneous cell populations in which the majority are T-cells, while our study focuses on intermediate and non-classical monocytes sorted in flow cytometry. A recent meta-analysis of scRNA-seq studies (Zernecke A et al. (2020, PMID: 32673538)) found monocytes form separate immune cell populations in mouse atherosclerotic aortas. Two mixed monocyte/DC subsets are well represented in all datasets with atherosclerosis, while few monocytes are present in healthy aortas. Of 559 marker genes of these monocyte-related cell subsets, 6 of them overlapped with our 90-gene NCM signature of comorbid HIV/CVD (CD2, LMNA, HOPX, GRASP, GPR35 and GPR171) and 2 of them overlapped with our 59-gene IM signature of comorbid HIV/CVD (PTGS2 and TRAF1). 

• Down this same track: Are the infected peripheral monocytes closer to monocytes in the plaque. The authors may use available monocyte transcriptomes from available scRNAseq studies.

Thanks for this interesting question. The single-cell RNA sequencing assisted in characterizing the heterogeneous nature of immune cell populations in atherosclerosis. A few scRNAseq studies investigated immune landscape of mouse and human atherosclerotic plaques (Cochain et al. (2018, PMID: 29545365), Fernandez DM et al.(2019, PMID: 31591603), Depuydt MAC et al.(2020, PMID: 32981416),Vallejo J et al. (2021, PMID: 34343272), Zernecke A et al.(2022, PMID: 36190844)). Not all of these studies provide publicly available gene signature of monocytes, though.

We compared our results to Cochain et al. (2018) who performed scRNA-seq on CD45+ cells from the aortas of non-diseased and atherosclerotic LDL receptor–deficient (Ldlr−/−) mice. Among the 13 cell subsets, Cochain et al. (2018) identified five myeloid cell subsets, including monocytes, monocyte-derived dendritic cells, and three macrophage subsets. We compared two comorbid HIV/CVD-associated gene signatures found in our study, with the marker gene lists for monocyte from Cochain et al. (2018) . 5 out of 59 genes in IIM signature overlapped with the reference gene list (NRIH3, PTGS2, IL18, DHX58 and ISG15), while 5 out of 90 genes in NCM signature overlapped with the reference gene list (NOCT, GPR35, GAS2L3, FAM20C and CSF3R). These overlapped genes were mainly related to cellular interaction, immune response, and inflammatory responses. 

In humans, Zernecke A et al. (2022) integrated scRNA-seq data from two studies. They recovered 10 clusters of cell subsets from a total of 2890 cells from 11 patients (n = 4 coronary vessels, n = 7 carotid endarterectomy specimens). Differential gene expression analyses across clusters identified cells corresponding to monocytes that are highly enriched for 116 marker genes. We examined the overlap between our comorbid HIV/CVD-associated gene signatures with these marker genes. Only 3 out of 59 genes in IIM signature overlapped with the reference gene signature (CCL4L2, CCL3 and PTGS2). None of 90 genes in NCM signature overlapped with the reference gene list. 

While we believe technical and experimental design reasons contribute to the discordance of our signatures and signatures from above scRNA-seq studies, different transcriptomic features and function of monocytes within plaque and in blood are important and require further studies.

• DE genes used as input for the pathway analysis are not evident from text or figures. These should be shown. Authors should label the corresponding genes in the volcano plots and explicitly state which genes were included in the pathway analysis, best as supplementary data file.

We thank the reviewer for this helpful comment. While we are not able to label all DE genes in the volcano plot due to limited space on the plot, we have added a reference for each corresponding DE gene lists, including their names and log fold changes, in the figure’s titles and the main text in context of the pathway analysis. 

Line 203: …in H+C+LLT- vs. H-C- comparison (S2 Table), using ingenuity pathway analysis.

Line 285: …IPA was conducted based upon 59 DEGs associated with the H+C+LLT- group (S2 Table) to characterize the gene transcription signal…

Line 339: …in H+C- or H+C+ vs. H-C- comparison (S4 Table), using ingenuity pathway analysis.

Line 342: …We conducted IPA of the 90-gene signature of H+C+ and the 37-gene signature of H+C- (S4 Table) to identify common or distinct elements (S3 Table B-C).

• E.g. Line 271 on page 7: "The role of these genes in IM was further investigated by ingenuity pathway analysis (IPA)..." and Line 277 page 7: "IPA was conducted to 59 DEGs associated with the H+C+LLT- group to characterize the gene transcription signal associated with comorbid HIV and CVD..." Which genes were used?

We greatly appreciate the reviewer for pointing out this omission through careful and detailed review of our manuscript. The first IPA analysis (Line 271) was for 22 LLT-associated DEGs, while the IPA analysis in the following paragraph (Line 277) was for 59 comorbid HIV/ CVD-associated DEGs.

To avoid confusion, we have clarified this on line 280.

Line 280: The role of these 22 LLT-associated DE genes in IM was further investigated by ingenuity pathway analysis (IPA).

• Line 250 on page 7: „By contrast, we detected an extensive DEG signature in IM associated with comorbid HIV/CVD (H+C+ versus H-C-, Fig 1B).“ In Figure 1B, no significantly regulated DEGs are shown for the comparison H+C+ vs H-C-. Line 299 on page 8: „As was found for IM, in NCM the H+C+ group had the largest set of DEGs….”. Please provide a list of genes.

Thank you for pointing out the need for more clarification. We have revised the text for clarity, which now reads (Line 258): By contrast, we detected an extensive DEG signature in IM associated with non-LLT users of comorbid HIV/CVD participants (H+C+LLT- versus H-C-, Fig 1B).

In addition, the information of 90 comorbid HIV/CVD associated DEG signature in NCM has been included in Supplementary table 4. We included a reference on Line 309.

Minor:

• The legends for the Volcano plots in Figures 1 & 2 are missing. I assume red dots are genes that show significant regulation.

Thank you for pointing out the need for clarification. Due to very limited space on the volcano plots, we clarified this in the title of Figure 1(Line 202) and Figure 2(Line 334): “Red genes indicate significance by FDR < 0.05 and |log2 fold-change |>1.”

• From the Methods section it becomes clear, that the authors performed DE Gene analysis in five comparisons including 6 groups (H+C-, H-C+, H+C+, H+C+LLT-, H+C+LLT+, H-C-). From this perspective, I find the schematic representation in Fig1a misleading, as only 4 groups are depicted. In addition, the orientation of the arrows towards the H-C- group is unclear. What is the intention of the authors?

Thank you for this helpful comment. The schematic design of our study focuses on three major comparisons of H+C-, H-C+ and H+C+ to healthy participants (H-C-). However, during analysis, we noticed a potential confounding effect of LLT use. Therefore, two additional contrasts of H+C+LLT- and H+C+LLT+ vs. H-C- were performed in both IM and NCM. 

Following your suggestion, we re-organized Figure 1A. Now it includes all five contrasts that have been studied, which is consistent with the method section. In addition, a label of “vs.” has been added near arrows to suggest the comparison group and the reference group of each contrast.

• In Figure 1b, some significantly regulated genes in the volcano plots are named while others are not (compare volcano plots for H+C+LLT- vs H-C- to H+C+LLT-+vs H-C-). Why were these genes labelled and others not? Presentation of data should be consistent.

• In line with the previous comment: l. 254 on page 7: "Up-regulated DEGs included known atherosclerosis genes such as the liver X receptor gene NR1H222, and NEXN, TRAF123, TLR7 and LGALS3BP24....". Please label in the volcano plot.

We thank the reviewers for this insightful comment. And we agree that labeling a few, but not all, DE genes on the volcano plots may bring some confusion. However, labeling hundreds of DE genes on these volcano plots is not practical due to the limited space. Therefore, we removed all DE gene labels from Figure 1B and Figure 2B. In addition, we add a reference of the DE gene information to the title of Figure 1B and 2B (Line 202 and Line 334).

Line202: … Red genes indicate significance by FDR < 0.05 and |log2 fold-change|>1(S2 Table).

Line 334: …Red genes indicate significance by FDR < 0.05 and |log2 fold-change |>1(S4 Table).

• The pathways shown in Figure 1D are not clearly associated with a particular group. Therefore, the authors are asked to add this information.

Thank you for pointing out this omission. A side label of “H+C+LLT-” has been added to Figure 1D to clarify this.

• I understand that DE genes with an adjusted p-value <0.05 were included in the pathway analysis. Were significant pathways detected by the adjusted or unadjusted p value? This is not clearly stated in the methods section. Unadjusted p-values are listed in Table S3. The authors are kindly requested to clarify this.

Thank you for pointing out this omission. The pathway significance was detected by the unadjusted p-value. We have now clarified our search method in the text as follows:

Lines 210: …Core analysis of IPA was subsequently performed and pathways and bio-functions with p-value≤0.05 were recognized as significant. 

• Please label Figure 3A&B with the corresponding models and refer to them correctly in the text (see line 359, page 9).

Thank you for pointing out the need for clarification. Figure 3A and 3B are now being labeled as “Salmon Module” and “Magenta Module”, respectively.

 

Response to Reviewer #2: 

This manuscript explores the difference in single cell transcriptome of Non Classical Monocytes and Intermediate Monocytes in PBMCs obtained from patient with HIV and CVD. The aim was to identified conserved differentially expressed gene signature in 92 patient divided in four groups according to their diagnosis (H-C- as a control, H+C-, H-,C+ and comorbidity of H+C+). This study is interesting because cardiovascular disease is one of the leading causes of mortality among people living with HIV and the use of PBMCs would be an affordable way to assess the risk and the mechanisms by which HIV engender CVD.

Comments:

1. The authors indicated that no difference in cell count was found, does this number refer to the quantification of the cells isolated from the CPT tube or the cells they isolated for sequencing? If the latter, can it really be said that there is no difference?

We thank the reviewers for this insightful comment. Identification and quantification of monocyte subtypes was done through flow cytometry analysis of PBMC samples. All monocytes were recovered from each PBMC sample. Then the relative frequency of each monocyte subtype was measured and compared across four HIV/CVD groups: H+C+, H+C-, H-C+ and H-C-. The sorted samples of each participant were frozen, and later, thawed to be used for subsequent RNA isolation and sequencing,

We have clarified this on line 157.

Line 157:…The quantification of intermediate and non-classical monocytes are estimated from the FACS data from all PBMC samples. 

2. In the results section, second paragraph. The values at the beginning are being presented in percentage, please keep it, it will be easy to correlate the 59% with undetectable RNA and 6 patients with higher RNA.

We greatly appreciate your careful and detailed review of our manuscript. This has now been clarified in the text on Line 237: “among these 44 women with HIV, only 14% had HIV RNA above 1000 copies/ml.” 

3. Do the authors looked at whether the 59% of HIV-positive patients who did not have RNA in plasma have any difference with patients with detected RNA and CDV? Wouldn't the presence or absence of RNA affect the behavior of peripheral cells, in this case monocytes? Or even if the disease is at a very advanced stage. This could perhaps give an idea why in the heatmap of the groups with IHV+ are not so homogeneous.

Thank you for the great suggestions. We agree that the presence of viremia is a factor that may affect the behavior of the monocytes. However, we were unable to compare groups of participants with versus without detected HIV-RNA, as the study was not designed for this comparison. Furthermore, participants in WIHS had relatively well controlled HIV disease, with access to effective therapy, so that the population may not have had sufficient variability in HIV disease severity to provide an informative comparison. We did include HIV disease stage related variables, including presence and absence of HIV RNA, among the many factors that were tested on the association with clustering pattern on the heat-map of DEGs. Unfortunately, we did not find any strong association between any of them to the expression pattern. Future studies will be needed to elucidate how these important HIV-related factors, including the amount of HIV viral load, duration of ART, advanced stage immunodeficiency and stage of HIV disease at diagnosis, etc., affect the differential gene expression of monocytes. We note this limitation on Line 475. 

 Line 475:… Future studies will be needed to elucidate how HIV-related factors, including the amount of HIV viral load, duration of ART and stage of HIV disease at diagnosis, etc., affect the differential gene expression of monocytes.

4. Heine GH, Rogacev KS had previously reported that the number of intermediate monocytes may be a significant predictor of cardiovascular events. What could be attributed in this case to the no difference found in monocytes between patients with CVD and negative for HIV?

Thank you for the thoughtful insight. We did notice that the results for comparison H-C+ vs. H-C- in our study are not consistent with a few prior studies. 

One possible reason is that the primary endpoint of Rogacev KS et al.(2012, PMID: 22999728) was clinical CVD events, which is defined as the first occurrence of cardiovascular death, acute myocardial infarction, or nonhemorrhagic stroke. In our study, we focused on the subclinical CVD, which is defined as the presence of carotid artery plaque evidenced by an area with localized IMT >1.5 mm in at least one of the six imaged artery locations. Although the vulnerable atheromatous plaque has been recognized as an important factor involved in the progression and regression of atherosclerosis, the correlation between reduction of atheromatous plaques with significant decreases in risk for acute cardiovascular events has not been established yet. 

In addition, our study has a relatively small sample size. Comparing to 951 subjects in Rogacev’s study, with around 22 participants in each group, we have a relatively low statistical power in detecting a difference in cell counts between patients with sCVD and negative for HIV (H-C+ vs. H-C-).

---

## [Editor Report · Decision Letter 1]

5 May 2023

HIV infection and cardiovascular disease have both shared and distinct monocyte gene expression features: Women's Interagency HIV Study

PONE-D-23-02143R1

Dear Dr. Kaplan,

We’re pleased to inform you that your manuscript has been judged scientifically suitable for publication and will be formally accepted for publication once it meets all outstanding technical requirements.

Kind regards,

Andreas Zirlik, MD

Academic Editor

PLOS ONE
---

## [Editor Report · Acceptance letter]

11 May 2023

PONE-D-23-02143R1 

HIV infection and cardiovascular disease have both shared and distinct monocyte gene expression features: Women's Interagency HIV Study 

Dear Dr. Kaplan:

I'm pleased to inform you that your manuscript has been deemed suitable for publication in PLOS ONE. Congratulations! Your manuscript is now with our production department. 

Kind regards, 

on behalf of

Univ. Prof. Dr. Andreas Zirlik 

Academic Editor

PLOS ONE